# Five New Cucurbitane-Type Triterpenoid Glycosides from the Rhizomes of *Hemsleya penxianensis* with Cytotoxic Activities

**DOI:** 10.3390/molecules24162937

**Published:** 2019-08-13

**Authors:** De-Li Chen, Xu-Dong Xu, Rong-Tao Li, Bo-Wen Wang, Meng Yu, Yang-Yang Liu, Guo-Xu Ma

**Affiliations:** 1Hainan Branch of Institute of Medicinal Plant Development, Chinese Academy of Medicinal Sciences & Peking Union Medical College (Hainan Provincial Key Laboratory of Resources Conservation and Development of Southern Medicine), Haikou 570311, China; 2Institute of Medicinal Plant Development, Chinese Academy of Medical Sciences & Peking Union Medical College, No. 151, Malianwa North Road, Haidian District, Beijing 100193, China; 3School of Chemical Engineering and Technology, Hainan University, Haikou 570228, China

**Keywords:** *Hemsleya penxianensis*, cucurbitane-typetriterpenoid glycosides, cytotoxicity

## Abstract

Five new cucurbitane-typetriterpenoid glycosides, named Xuedanoside F–J (**1**–**5**), were obtained from the rhizomes of *Hemsleya penxianensis* (Xue dan), which belongs to the family of Cucurbitaceae. These new compounds were elucidated byspectroscopic analysis, including 1D, 2D NMR, and HR-ESI-MS spectra. Additionally, all the isolates were evaluated for cytotoxicity against three human cancer cell lines (Hela, MCF-7, and A-549) with the IC_50_ ranging from 2.25 to 49.44 µM in vitro with treatment 48 h and showed low cytotoxicity in human normal liver L-02 cells (IC_50_ > 50 µM). Compound **5** showed the most significant cytotoxic activity with the IC_50_ value of 2.25, 4.72, and 5.33 µM in 48 h, respectively.

## 1. Introduction

*Hemsleya pengxianensis* W.J. Chang, a native plant andwidely distributed in the south-west provinces of China, belongs to the genus of *Hemsleya* in Cucurbitaceae family [1]. It is also called “Xue dan” dialectally and has been used as a traditional Chinese medicine for a long time [2]. The tubers of *H. pengxianensis* have been dispensed for a variety of ailments including bacillary dysentery, sore throats, stomachaches, toothaches, diarrhea, ulcers, jaundice, bronchitis, chronic cervicitis, and tuberculosis [2,3,4].

Previous phytochemical reports have indicated that *Hemsleya* spp.possess rich terpenoid compounds including diterpenes, triterpenes, and particularly cucurbitane triterpenoid glycosides, which are efficient in the cureof all kinds of inflammation and cancers [5,6,7,8,9,10]. In prior research, our studies led to the disclosure of nineteen new cucurbitane-type triterpenoids that have shown significant anti-tumor cytotoxicity [11,12,13]. Recently, further study of *H. pengxianensis* has found another five new cucurbitane triterpenoid saponins named Xuedanoside F-J (**1**–**5**) (Figure 1), which were isolated from the rhizomes of *H. penxianensis*. In this paper, we report the isolation and structure identification of Xuedanoside F-J and evaluate their cytotoxic activity against human cancer cell lines.

## 2. Results

Compound **1** was isolated as an amorphous white powder with ${\left[\alpha \right]}_{D}^{30}$ +60.5 (c 0.1, MeOH). The molecular formula was determined as C_36_H_56_O_11_ according to the molecular ion peak at *m*/*z* [M + Na]^+^ 687.3725in the HR-ESI-MS (calculated for 687.3720 C_36_H_56_NaO_11_). Its IR data displayed absorptions for hydroxyl (3565–3340 cm^-1^) and carbonyl (1651 and 1687 cm^-1^) groups. Acid hydrolysis of **1** with HCl gave D-glucose as the constituent unit, which was tested by GC analysis. D-Glucose (t*_R_* = 25.5 min) was detected by comparison with authentic monosaccharide. The configuration of the glycosidic bond was β on the basis of the coupling constant of the anomeric proton at *δ*_H_ 4.80 (d, *J* = 6.0 Hz). The ^1^H-NMR data (Table 1) revealed the existence of seven angular methyl signals at *δ*_H_ 1.26 (s), 1.21 (s), 1.44 (s), 1.91 (s), 1.34 (s), 1.28 (s), and 1.45 (s); four oxygenated methines at *δ*_H_4.08 (m), 4.20 (m), 3.40 (d, *J* = 12.0 Hz), 5.06 (m), and 5.19 (t, *J* = 6.0 Hz); and two olefinic proton signals at *δ*_H_ 5.68 (m) and 6.89 (d, *J* = 6.0 Hz). The ^13^C APT NMR data (Table 2) showed 36 carbon signals due to 7 methyls (δ_C_21.4, 21.8, 23.2, 22.4, 23.6, 26.7, 31.4), 6 methylenes, 9 methines and 8 quaternary carbons (including 2 olefinic carbon at δ_C_ 143.7, 135.7 and one carbonyl carbons at δ_C_ 214.4), of which 30 were assigned to the aglycon part, and the remaining 6 were ascribed to the sugar moiety. All assignments of proton signals achieved by ^1^H and ^13^C correlations in the HSQC spectrum. The IR and ^1^H and ^13^C-NMR spectra data identified that compound **1** is an oxygenated cucurbitane triterpenoid glycoside derivative [11]. The connectivities of compound **1** were deduced mainly by ^1^H-^1^H COSY and HMBC spectra (Figure 2). Analysis of the HMBC spectrum (Appendix A), the correlations from δ_H_3.40 (H-3) to δ_C_ 72.2 (C-2) and δ_C_ 44.1 (C-4), and δ_H_ 2.95 (H-17) to δ_C_ 80.8 (C-20) suggested the presence of hydroxyl groups at C-2, C-3, and C-20, respectively. Besides, HMBC correlations of H-6 with C-5 (δ_C_ 143.7) and C-7 (δ_C_ 22.5), H-24 with C-23 (δ_C_ 71.8), C-25 (δ_C_ 135.7), and C-27 (δ_C_ 23.2), H-12 with C-11 (δ_C_ 214.4) implied that olefinic groups were at C-5 and C-25, and a carbonyl group was at C-11. Comprehensive comparison of the NMR data of **1** with those of the known compound hemslelis A [10] suggested that compound **1** was an analogue of hemslelis A, except that compound **1** contained one D-glucose and lost a carbonyl group at C-7. The location of the sugar unit was located at C-26 by an O atom due to the HMBC correlations (Figure 2) from the proton signal at δ_H_ 4.85 (H-26) to anomeric carbon at δ_C_ 103.9, and the signal for C-26 revealed a powerful downfield shift to δ 68.0 (+6.8 ppm). In the NOESY spectrum (Appendix A), correlations from H-2 to H-10, H-3 to H-19 indicated that OH-2 was β-oriented, and OH-3 was α-oriented, respectively. Furthermore, the ^3^*J* coupling constant (*J* = 12.0 Hz) verified the antiperiplanar link between H-2 and H-3. NOESY correlations from H_3_-18 to H-16 corroborated that these protons were inthe β-orientation, and the coupling constant (*J* = 12.0 Hz) also supported the antiperiplanar relationship between H-16 and H-17. The six-member ring through O atom between C-16 and C-23 suggested the synperiplanar conformation of H-16 and H-23. Therefore, taken along with ^1^H-^1^H COSY, HSQC, HMBC, and NOESY spectra (Appendix A), the structure of compound **1** was established as 2β, 3α, 20β-trihydroxycucurbita-16α-23α-epoxy-5, 24(E)-diene-11-one-26-O- D-glucopyranoside and it was named Xuedanoside F.

Compound **2** was obtained as a shapeless white powder with ${\left[\alpha \right]}_{D}^{24}$+ 83.8 (*c* 0.1, MeOH). Its molecular formula was established as C_36_H_56_O_10_ based on its HR-ESI-MS spectrum at *m*/*z* [M + Na]^+^ 671.3768 (calculated for C_30_H_46_NaO_4_, 671.3771). An analysis of the ^1^H and ^13^C-NMR data (Table 1 and Table 2) displayed that the structure of **2** was similar to that of **1**. An unambiguous comparison the data of **2** with **1** shown that oxymethine at C-2 in **2** was absent. Furthermore, it was observed that the carbon signal at C-3, in comparison with **1**, evidently shifted to δ_C_ 76.0 (–12.7 ppm) in ^13^C-NMR data of **2**. Additionally, in the HMBC spectrum (Appendix A), correlations from H-2 to C-4 proved the deficiency of the group. The significant NOESY (Appendix A) correlations from H-10 (δ_H_ 2.54) to H-3 (δ_H_ 3.70), from H-3 (δ_H_ 3.70) to H_3_-29 (δ_H_ 1.13) confirmed the relative configurations of methyl groups and other protons in the tetracyclic rings. The coupling constant of *J* = 12.0 Hz further confirmed the antiperiplanar relationship between H-16 and H-17.Taken together with the analysis of NOE spectra between the two compounds, compound **2** was elucidated as 3β, 20β-dihydroxycucurbita-16α-23α-epoxy-5, 24(E)-diene-11-one-26-O-d- glucopyranoside and it was named Xuedanoside G.

Compound **3** was determined to be a molecular formula of C_36_H_58_O_8_, as established with its HR-ESI-MS data at *m*/*z* [M + Na]^+^ 641.4021 (calculated for C_36_H_58_NaO_8_, 641.4029). Comparing its ^1^H and ^13^C-NMR data (Table 1 and Table 2) with that of **2** showed that their structures were close, with the exception of the presence of the sugar group at C-3 (δ_C_ 87.7) in **3** instead of the sugar group at C-26 (δ_C_ 67.1) in **2**. Similarly, itlackedthe a hydroxyl group at C-20 (δ_C_ 36.4) and the loss of an ether bond between H-16 and C-23 in **3**. In the HMBC spectrum (Appendix A), the sugar unit was linked at C-3 according to the correlation from the proton signal at δ_H_ 3.62 (H-3) to anomeric carbon at δ_C_ 107.8 (C-1′), and the signal for C-3 indicated a significant downfield shift to δ 87.7 (+11.7 ppm). Similarly, in comparison to **2**, the signals for C-16, C-20, and C-23 revealed the powerful upfield shift to δ 28.5 (–42.7 ppm), δ 36.4 (–36.6 ppm), and δ 25.1 (–45.9 ppm), respectively, while a hydroxyl group and an ether bond were absent. Compound **3** was eventually determined to be 26-hydroxycucurbita-5, 24(E)-diene-11-one-3-O-β-d-glucopyranoside, and it was named Xuedanoside H.

Compound **4** had a molecular formula C_36_H_60_O_8_ (Calcd for C_36_H_60_NaO_8_, 643.4186) on the basis of ion peak at *m*/*z* [M + Na]^+^ 643.4178 in HR-ESI-MS. The 1D NMR signals (Table 1 and Table 2) were tightly connected to those of **3**, with the difference of the carbonyl group of C-11 (δ_C_ 214.2) in **3**, where it was substituted for a hydroxy group at δ_C_ 78.6 in compound **4**. This difference was verified by 2D NMR spectra (Appendix A). In the HMBC spectrum, the correlations from H-11 at δ_H_ 4.18 to C-8 (δ_C_ 43.9), C-10 (δ_C_ 37.3), and C-13 (δ_C_ 47.8) revealed that the hydroxyl group was located at C-11. Within the NOESY spectrum, from H_3_-19 to H-11 and from H-11 to H_3_-18 suggested that H-11 was β-oriented, and the structure of compound **4** was established as 11α, 26-dihydroxycucurbita-5, 24(E)-diene-3-O-β-d-glucopyranoside, and it was named Xuedanoside I.

Compound **5** possesses a molecular formula of C_36_H_56_O_10_on the basis of HR-ESI-MS at *m/z* [M + Na]^+^ 671.3779 (calculated for C_36_H_56_NaO_10_, 671.3771) and NMR spectra. Its ^1^H and ^13^C-NMR (Table 1 and Table 2) data are similar to those of compound **3**, with the exception of the addition of a carbonyl group at C-7 (δ_C_ 199.6) and a hydroxyl group at C-27 (δ_C_ 58.9) in **5**, respectively. In the HMBC spectrum (Appendix A), the correlations of H-6 (δ_H_ 6.31) with the downfield carbon C-7 (δ_C_ 199.6) (compared with C-7 in **3**) implied acarbonyl group at C-7. Furthermore, in comparison to **3**, the signal for C-27 revealed a powerful downfield shift to δ 58.9 (+44.4 ppm), while a hydroxyl group was added at C-27. The form of **5** was confirmed by spectra of ^1^H-^1^H COSY, HSQC, HMBC, and NOESY (Appendix A); it wasidentified as 26, 27-dihydroxycucurbita-5,24(E)- diene-7,11-dione-3-O-β-d-glucopyranoside and was named Xuedanoside J.

Furthermore, the cytotoxicity of all isolates was assessed with three human tumor cell lines (Hela, MCF-7, and A-549) according to the MTT procedure, and doxorubicin was used as the positive control. The results of cytotoxicity were displayed in Table 3. Compound **5** exhibited remarkable cytotoxicity against Hela, MCF-7, and A-549 cell lines with IC_50_ values from 2.25 to 5.33 μM in 48 h. Compounds **3**–**4** showed moderate cytotoxicity with the IC_50_ values between 7.55 and 18.72 μM in 48 h, whereas compounds **1** and **2** had weak effects with IC_50_> 30 μM. Meanwhile, the results revealed that tested compounds had low cytotoxic activity with the IC_50_ value more than 50.0 μM in normal human liver L-02 cells when compared to the control drug, doxorubicin (IC_50_ = 15.42 μM).

## 3. Discussion

Cucurbitane triterpene and its glycoside derivatives widely exist in the genus of *Hemsleya*, which are the effective constituents and show potent anti-tumor cytotoxicity. As a result, we evaluated all the isolates for their cytotoxic activity against three human cancer cell lines. Compared to the doxorubicin positive control group, all compounds showed moderate cytotoxicity due to their 24-ethylenic linkage substituent [8], with the value of IC_50_ ranging from 2.25 to 49.44 µM. Compound **5** displayed the most significant cytotoxic activity, which may be related to the carbonyl group at C-7 as a characteristic structural unit compared to its derivatives. Compounds **1** and **2** revealed the weak cytotoxic activity when compared with the other isolates, which may be caused by the formation of ether bond between C-16 and C-23. In brief, the A ring and branch chain had dramatic effects on potency against human tumor cell lines. All compounds showed low cytotoxic activity in human normal liver L-02 cells when compared to doxorubicin. Based on these promising results, compounds **3** and **5** could serve as potential anti-cancer agents for future cancer chemotherapy.

## 4. Materials and Methods

### 4.1. General Experimental Procedures

1D and 2D NMR spectra were obtained with a Bruker AV 600 NMR spectrometer(chemical shifts are presented asδ values with TMS as the internal standard) (Bruker, Billerica, Germany). HR-ESI-MS were performed on a Q-tof spectrometer (Waters, Milford, MA, USA). UV and IR data were done using a Shimadzu UV2550 and FTIR-8400Sspectrometer (Shimadzu, Kyoto, Japan), respectively. Thin-layer chromatography (TLC) was performed on pre-coated silica gel GF_254_ (Zhi Fu Huang Wu Pilot Plant of Silica Gel Development, Yantai, China). Semi-preparative HPLCwas conducted on an analytic LC equipped with a pump of P230, a DAD detector of 230+ (Ellte, Dalian, China) with a C_18_ ODS-A (5 µm, YMC, Kyoto, Japan). Column chromatography with silica gel was used (100-200 and 200-300 mesh, Qingdao Marine Chemical plant, Qingdao, China). All solvents used were of analytical grade (Beijing Chemical Plant, China).

### 4.2. Plant Material

The rhizomes of *Hemsleya penxianensis* (Cucurbitaceae) were collected in the Jinfuo mountain, Nanchuan district of Chongqing City, China, on September 2014, and were identified by Prof. Si-Rong Yi, Institute of Medicinal Plant Development, Chinese Academy of Medical Sciences, where the voucher specimen (CS140921) was stored. The plant drug was dried in the shade, powdered, and contained in an airtight container.

### 4.3. Extraction and Isolation

The rhizomes of *H. penxianensis* (10.0 kg) were extracted with 95% EtOH under reflux (3 h × 60 L × 3). The EtOH extract was evaporated at 50 °C, and the crude extracts were dissolved in water. The aqueous extraction was re-extracted with EtOAc, and an EtOAc fraction was obtained. The fraction of EtOAc (200 g) was subjected to silica gel column chromatography and eluted with a gradient system of CH_2_Cl_2_-MeOH to obtain 12 fractions (Fr. A-Fr. L).

The fraction J (16.3 g) was subjected to column chromatography on silica gel and eluted with CH_2_Cl_2_-MeOH gradient (60:1, 40:1, 30:1, 20:1, 10:1, 5:1 *v*/*v*), to obtain 6 fractions (Fr. I-VI). The Fr. IV (3.2 g) was further separated by MCI-gel column chromatography with methanol-water (10:90, 20:80, 30:70, 40:60, 50:50, 70:70, 90:10, 100:0) gradient elution, giving 8 fractions (Fr. IV.1–IV.8). Fraction IV.3 was subjected to semi-preparative HPLC with CH_3_CN-H_2_O as the mobile phase (18:82, v/v) by the YMC-Pack ODS-A column to acquire compound **1** (8.7 mg, t*_R_* = 12.4 min) and **2** (8.8 mg, t*_R_* = 17.8 min). Fraction IV.4 was prepared by semi-preparative HPLC eluting with CH_3_CN-H_2_O (16:84, *v*/*v*) to give compound **3** (6.7 mg, t*_R_*= 15.2 min), **4** (9.5 mg, t*_R_* = 22.8 min), and **5** (8.7 mg, t*_R_* = 26.4 min).

The structures of compounds **1**-**5** were determined by HR-ESI-MS, UV, IR, 1D, and 2D NMR spectra.

*Xuedanoside F* (**1**). C_36_H_56_O_11_,${\left[\alpha \right]}_{D}^{30}\;$+ 60.5 (c 0.1, MeOH), white amorphous powder; IR (KBr) ν_max_ cm^-1^: 1651, 1687, 3565-3340; UV *λ*_max_ (MeOH) nm (log *ε*): 205.8 (5.80); HR-ESI-MS *m/z* [M + Na]^+^ 687.3725(calcd. 687.3720); ^1^H and ^13^C-NMR spectra data, see Table 1 and Table 2.

*Xuedanoside G* (**2**). C_36_H_56_O_10_, ${\left[\alpha \right]}_{D}^{24}\;$+ 83.8 (*c* 0.1, MeOH), white amorphous powder; IR (KBr) ν_max_ cm^-1^: 1675, 1689, 3569-3254; UV *λ*_max_ (MeOH) nm (log *ε*): 210.5 (5.68); HR-ESI-MS *m*/*z* [M + Na]^+^ 671.3768(calcd. 671.3771); ^1^H and ^13^C-NMR spectra data, see Table 1 and Table 2.

*Xuedanoside H* (**3**). C_36_H_58_O_8_, ${\left[\alpha \right]}_{D}^{27}\;$+ 69.7 (*c* 0.1, MeOH), white amorphous powder; IR (KBr) ν_max_ cm^-1^: 1170, 1661, 1723, 3633-3354; UV *λ*_max_ (MeOH) nm (log *ε*): 202.8 (5.10); HR-ESI-MS *m*/*z* [M + Na]^+^ 641.4021(calcd. 641.4029); ^1^H and ^13^C-NMR spectra data, see Table 1 and Table 2.

*Xuedanoside I* (**4**). C_36_H_60_O_8_, ${\left[\alpha \right]}_{D}^{21}\;$+ 30.9 (*c* 0.1, MeOH), white amorphous powder; IR (KBr) ν_max_ cm^-1^: 1145, 1640, 3658-3385; UV *λ*_max_ (MeOH) nm (log *ε*): 202.4 (5.08); HR-ESI-MS *m*/*z* [M + Na]^+^ 643.4178 (calcd.643.4186); ^1^H and ^13^C-NMR spectra data, see Table 1 and Table 2.

*Xuedanoside J* (**5**). C_36_H_56_O_10_, ${\left[\alpha \right]}_{D}^{27}\;$+ 59.7 (*c* 0.1, MeOH), white amorphous powder; IR (KBr) ν_max_ cm^-1^: 1195, 1651, 1680, 3650-3460; UV *λ*_max_ (MeOH) nm (log *ε*): 209.6 (5.60); HR-ESI-MS *m*/*z* 671.3779 [M + Na]^+^ (calcd.671.3771); ^1^H and ^13^C-NMR spectra data, see Table 1 and Table 2.

Acid hydrolysis of Compounds **1**–**5** were accomplished by the procedure described previously [14,15].

### 4.4. Cytotoxicity Assays

Compounds **1**–**5** isolated from *H. penxianensis* were screened for cytotoxicity against three human cancer cell lines, including Hela, breast cancer MCF-7, lung cancer A-549, and the normal liver L-02 cells. This used the MTT method as described in previously published literature [16,17]. Briefly, the cells, at a density of 1.1 × 10^5^ cells/mL in 96-well microtiter plate, were cultured in DMEM medium with 10% fetal bovine serum at 37 °C in a 5% CO_2_ incubator overnight. Then, the cells were treated with the test compounds at five concentrations in triplicate. After 24 h and 48 h of treatment, the cells were incubated with 10 μL of MTT (4 mg/mL) for another 4 h. The residual liquid was removed, and 200 µL DMSO was added. The absorbance was tested using a microplate reader at a wavelength of 570 nm.

## Figures and Tables

**Figure 1 molecules-24-02937-f001:**
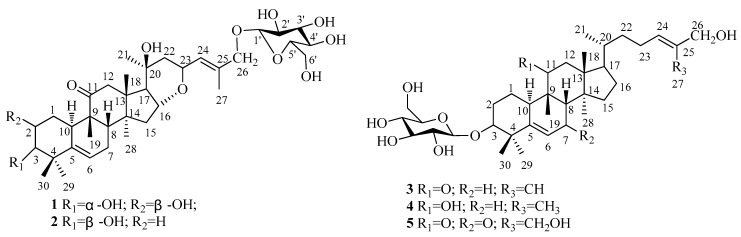
Structures of compounds **1**–**5**.

**Figure 2 molecules-24-02937-f002:**
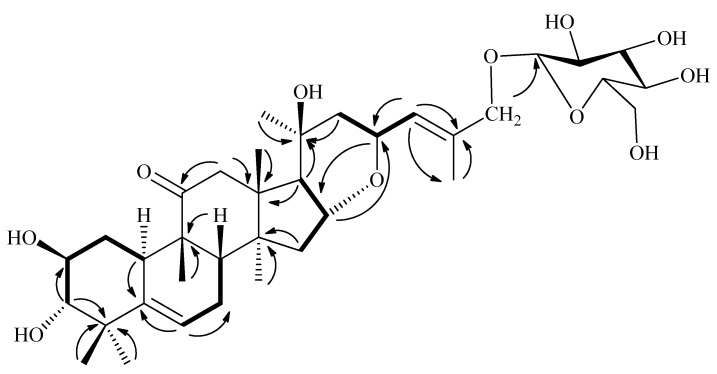
^1^H-^1^H COSY and HMBC correlations of compounds **1** (
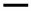

^1^H-^1^H COSY; 
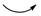
 HMBC).

**Table 1 molecules-24-02937-t001:** ^1^H-NMR Spectra Data (600 MHz, pyridine-*d*_5_) for Compounds **1**–**5** (δ_H_ in ppm, *J* in Hz).

Position	1	2	3	4	5
1	1.52 (1H, m) 2.44 (1H, m)	2.07 (1H, m) 1.72 (1H, m)	1.75 (1H, m) 1.67 (1H, m)	2.92 (1H, m) 2.01 (1H, m)	2.06 (1H, m) 1.62 (1H, m)
2	4.08 (1H, m)	2.05 (1H, m) 1.89 (1H, m)	2.37 (1H, m) 1.90 (1H, m)	2.41 (1H, m) 2.02 (1H, m)	1.85 (1H, m) 2.38 (1H, m)
3	3.40 (1H, d, 12.0)	3.70 (1H, s)	3.62 (1H, s)	3.67 (1H, s)	3.70 (1H, s)
6	5.68 (1H, m)	5.66 (1H, d, 12.0)	5.52 (1H, d, 6.0)	5.49 (1H, d, 6.0)	6.31 (1H, s)
7	1.85 (1H, m) 2.28 (1H, m)	2.24 (1H, m) 1.78 (1H, m)	1.81 (1H, m) 1.94 (1H, m)	2.29 (1H, m) 1.70 (1H, m)	
8	1.93 (1H, m)	1.84 (1H, m)	1.80 (1H, m)	1.62 (1H, m)	2.62 (1H, s)
10	2.66 (1H, m)	2.54 (1H, d, 14.4)	2.47 (1H, m)	2.79 (1H, d, 10.8)	2.98 (1H, m)
11				4.18 (1H, m)	
12	2.64 (1H, m) 3.17 (1H, d, 12.0)	3.21 (1H, d, 14.4) 2.68 (1H, d, 14.4)	2.49 (1H, m) 2.94 (1H, d, 12.0)	2.12 (1H, m) 2.07 (1H, m)	2.94 (1H, d, 18.0) 2.52 (1H, d, 12.0)
15	1.61 (1H, m) 1.92 (1H, m)	1.90 (1H, m) 1.62 (1H, m)	1.30 (1H, m) 1.38 (1H, m)	1.24 (1H, m) 1.07 (1H, m)	1.40 (1H, m) 1.80 (1H, m)
16	5.06 (1H, m)	5.21 (1H, t, 6.0)	1.27 (1H, m) 2.13 (1H, m)	1.87 (1H, m) 1.18 (1H, m)	1.26 (1H, m) 1.88 (1H, m)
17	2.14 (1H, d, 12.0)	2.16 (1H, d, 9.0)	1.68 (1H, m)	1.61 (1H, m)	1.64 (1H, m)
18	1.26 (3H, s)	1.27 (3H, s)	0.70 (3H, s)	0.89 (3H, s)	0.68 (3H, s)
19	1.21 (3H, s)	1.24 (3H, s)	1.14 (3H, s)	1.31 (3H, s)	1.12 (3H, s)
20			1.45 (1H, m)	1.58 (1H, m)	1.38 (1H, m)
21	1.44 (3H, s)	1.45 (3H, s)	0.89 (3H, s)	0.94 (3H, s)	0.80 (3H, s)
22	1.79 (1H, m) 2.07 (1H, q, 6.0)	2.09 (1H, m) 1.81 (1H, m)	1.52 (1H, m) 1.18 (1H, m)	1.45 (1H, m) 1.09 (1H, m)	1.17 (1H, m) 1.58 (1H, m)
23	5.19 (1H, t, 6.0)	5.11 (1H, m)	2.18 (2H, m)	2.11 (1H, m) 1.98 (1H, m)	2.16 (1H, m) 2.30 (1H, m)
24	6.89 (1H, d, 6.0)	6.92 (1H, d, 9)	5.72 (1H, t, 6.0)	5.65 (1H, t, 7.2)	5.88 (1H, t, 6.0)
26	4.85 (1H, d, 6.0) 4.44 (1H, d, 6.0)	4.86 (1H, d, 12.0) 4.45 (1H, d, 12.0)	4.31 (2H, s)	4.32 (2H, s)	4.73 (2H, s)
27	1.91 (3H, s)	1.91 (3H, s)	1.83 (3H, s)	1.80 (3H, s)	4.70 (2H, s)
28	1.34 (3H, s)	1.37 (3H, s)	0.98 (3H, s)	0.91 (3H, s)	1.05 (3H, s)
29	1.28 (3H, s)	1.13 (3H, s)	1.10 (3H, s)	1.15 (3H, s)	1.18 (3H, s)
30	1.45 (3H, s)	1.41 (3H, s)	1.54 (3H, s)	1.56 (3H, s)	1.58 (3H, s)
Glc					
1′	4.80 (1H, d, 6.0)	4.81 (1H, d, 7.8)	4.83 (1H, d, 6.0)	4.91 (1H, d, 7.8)	4.86 (1H, d, 6.0)
2′	4.02 (1H, m)	4.05 (1H, m)	3.95 (1H, m)	3.98 (1H, m)	3.97 (1H, m)
3′	4.17 (1H, m)	4.22 (1H, m)	4.18 (1H, m)	4.21 (1H, m)	4.21 (1H, m)
4′	4.18 (1H, m)	4.21 (1H, m)	4.16 (1H, m)	4.21 (1H, m)	4.20 (1H, m)
5′	3.89 (1H, m)	3.95 (1H, m)	3.92 (1H, m)	3.93 (1H, m)	3.95 (1H, m)
6′	4.55 (1H, d, 6.0) 4.35 (1H, m)	4.58 (1H, d, 12.0) 4.39 (1H, m)	4.50 (1H, d, 12.0) 4.35 (1H, m)	4.57 (1H, d, 12.0) 4.41 (1H, m)	4.55 (1H, d, 12.0) 4.40 (1H, m)

**Table 2 molecules-24-02937-t002:** ^13^C-NMR (150MHz, pyridine-*d_5_*) spectral data of compounds **1**–**5**.

Position	1	2	3	4	5
1	35.9	21.6	22.6	27.2	22.5
2	72.2	30.3	28.8	30.0	28.4
3	82.7	76.0	87.7	88.3	87.2
4	44.1	42.4	42.5	42.8	43.9
5	143.7	141.9	141.7	144.7	168.3
6	120.0	119.4	118.9	118.9	125.4
7	25.5	24.7	24.6	25.0	199.6
8	44.1	43.5	44.4	43.9	60.0
9	50.50	50.2	49.5	40.5	49.5
10	35.5	36.1	36.4	37.3	38.0
11	214.4	213.8	214.2	78.6	211.7
12	50.1	49.4	49.2	41.5	49.1
13	50.0	49.3	49.4	47.8	48.9
14	49.9	49.2	50.0	50.1	49.7
15	42.9	42.2	35.0	34.9	35.2
16	72.1	71.2	28.5	28.7	28.2
17	57.3	56.5	50.1	51.0	49.6
18	21.4	20.5	17.4	19.2	17.4
19	21.8	20.8	20.8	26.7	21.3
20	73.8	73.0	36.4	36.5	36.3
21	31.4	30.6	18.7	19.7	18.8
22	47.7	47.0	36.8	37.5	37.0
23	71.8	71.0	25.1	25.2	24.8
24	133.5	132.7	125.4	127.7	127.7
25	135.7	134.8	136.7	136.7	141.3
26	68.0	67.1	68.5	68.5	65.8
27	23.2	22.4	14.5	17.4	58.9
28	22.4	21.6	19.0	28.1	18.9
29	23.6	28.2	28.9	26.8	28.6
30	26.7	26.7	26.3	22.3	25.6
Glc					
1′	103.9	103.0	107.8	107.8	107.6
2′	76.3	75.6	75.9	75.9	75.9
3′	79.9	79.1	79.1	79.1	79.1
4′	73.0	72.2	72.1	72.2	72.1
5′	79.8	79.0	78.6	78.2	78.8
6′	64.1	63.3	63.4	63.4	63.4

**Table 3 molecules-24-02937-t003:** Cytotoxicity (IC_50_, μM ± SD) of compounds **1**–**5** against three human cancer cell lines.

Compounds	Hela		MCF-7		A-549		L-02	
	48 h	24 h	48 h	24 h	48 h	24 h	48 h	24 h
**1**	34.38±2.05	50.56 ± 4.28	45.09 ± 3.52	57.85 ± 5.16	49.44 ± 2.67	68.82 ± 4.33	>100	>100
**2**	31.75 ± 1.45	40.32 ± 2.56	45.88 ± 0.92	60.74 ± 4.73	47.58 ± 0.84	80.65 ± 5.16	>100	>100
**3**	7.55 ± 1.75	13.15 ± 1.88	10.88 ± 2.77	26.12 ± 1.22	8.55 ± 1.78	20.12 ± 1.08	68.25 ± 3.78	>100
**4**	14.77 ± 2.15	25.38 ± 3.72	12.54 ± 1.32	25.44 ± 3.15	18.72 ± 2.35	40.18 ± 3.02	89.55 ± 4.60	>100
**5**	2.25 ± 0.42	4.88 ± 1.05	4.72 ± 0.54	12.65 ± 2.36	5.33 ± 0.68	12.45 ± 1.28	50.52 ± 2.15	>100
doxorubicin	1.32 ± 0.03	2.15 ± 0.06	2.45 ± 0.05	3.02 ± 0.04	3.85 ± 0.05	6.10 ± 0.26	15.42 ± 0.28	26.56 ± 1.35

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
