# Peer review of "Five New Cucurbitane-Type Triterpenoid Glycosides from the Rhizomes of Hemsleya penxianensis with Cytotoxic Activities"

_molecules, 2019, doi:10.3390/molecules24162937_

Round 1

Reviewer 1 Report

Although the authors carried out the required revisions on the manuscript, some minor corrections are still needed.

In the abstract, language polishing is needed, since the authors only added information without language concerns.

- “triterpenoids glycosides” should be replaced by “triterpenoid glycosides”.

- The last sentence of the abstract should be split into two. Rather than just telling that compound 5 was the most active one, which, by just reading the abstract, does not give any information about the compound’s structure or its activity, the effect of this compound should be highlighted, e.g. by mentioning that it showed IC50 values in the single digit micromolar concentration against all cancer cells. “especially” in the end of the sentence should be deleted.

Since the results on the cytotoxicity evaluation revealed a higher activity of the compounds to cancer cells relatively to normal ones, with selectivity indexes comparable or higher than those of doxorubicin, rather than just mentioning “that tested compounds had moderate cytotoxic activity in human normal liver L-02 cells”, the selectivity to cancer cells should be highlighted in the abstract as well as in the results and discussion. Moreover, rather than moderate, the cytotoxicity to normal cells can be considered low.

Reviewer 2 Report

the authors adressed most of the comments but they need to add only the 24 hr treatment results if possible.
